# Effects of High Salt Intake on Detrusor Muscle Contraction in Dahl Salt-Sensitive Rats

**DOI:** 10.3390/nu13020539

**Published:** 2021-02-07

**Authors:** Ryoya Kawata, Yuji Hotta, Kotomi Maeda, Tomoya Kataoka, Kazunori Kimura

**Affiliations:** 1Department of Hospital Pharmacy, Nagoya City University Graduate School of Pharmaceutical Sciences, Nagoya 467-8603, Japan; r.k.1228q@gmail.com (R.K.); 5x10x3@gmail.com (K.M.); 2Department of Clinical Pharmaceutics, Nagoya City University Graduate School of Medical Sciences, Nagoya 467-8601, Japan; kataoka@med.nagoya-cu.ac.jp

**Keywords:** lower urinary tract symptoms, salt-sensitivity, detrusor muscle, rats

## Abstract

High salt intake has been reported as a risk factor for urinary storage symptoms. However, the association between high salt intake and detrusor muscle contraction is not clear. Therefore, we investigated the effects of high salt intake on the components of detrusor muscle contraction in rats. Six-week-old male Dahl salt-resistant (DR; *n* = 5) and Dahl salt-sensitive (DS; *n* = 5) rats were fed a high salt (8% NaCl) diet for one week. The contractile responses of the detrusor muscle to the cumulative administration of carbachol and electrical field stimulation (EFS) with and without suramin and atropine were evaluated via isometric tension study. The concentration–response curves of carbachol were shifted more to the left in the DS group than those in the DR group. Contractile responses to EFS were more enhanced in the DS group than those in the DR group (*p* < 0.05). Cholinergic component-induced responses were more enhanced in the DS group than those in the DR group (*p* < 0.05). High salt intake might cause urinary storage symptoms via abnormalities in detrusor muscle contraction and the enhancement of cholinergic signals. Excessive salt intake should be avoided to preserve bladder function.

## 1. Introduction

In modern aging societies, lower urinary tract symptoms (LUTS) are a serious public health concern [1]. LUTS refers to symptoms related to urination, which can be divided into urinary storage symptoms, voiding symptoms, and post-micturition symptoms [2]. LUTS significantly reduce the quality of life by limiting the activities of daily living [3]. As bladder dysfunction that leads to LUTS may have different causes, prevention and treatment methods that consider the patient’s underlying condition are needed [4]. Despite the availability of a range of treatment methods, including behavioral, pharmacological, and surgical approaches, many cases of LUTS are refractory [5,6,7]. Previous epidemiological studies reported that lifestyle habits such as overeating, smoking, and stress are associated with the development and progression of urinary storage symptoms, including urinary frequency and urgency [8,9,10]. A recent cross-sectional study conducted in Japan reported that high salt intake is one of the causes of urinary storage symptoms such as nocturia and nocturnal polyuria [11].

Moreover, other previous studies reported the relationship between high salt intake and urinary storage symptoms [12,13]. Dahl salt-sensitive (DS) and Dahl salt-resistant (DR) rats are widely used for basic studies on the relationship between salt intake and disease [14]. We previously reported that one week of high salt intake in DS rats was associated with the upregulation of epithelial sodium channel alpha expression and contributed to urinary frequency [12]. These results suggest that high salt intake in DS rats induces urinary storage symptoms after short-term exposure. However, the association between high salt intake and detrusor muscle contraction is not clear.

The detrusor muscle is a smooth muscle present in the bladder wall that regulates urinary storage and voiding. The parasympathetic nerves predominate the control of the detrusor muscle, which contracts during voiding [15]. The contraction of the detrusor muscle is caused by neurotransmitters released from the parasympathetic postganglionic nerve endings. The components of detrusor muscle contraction can be divided into purinergic and cholinergic components with adenosine 5′-triphosphate (ATP) and acetylcholine as neurotransmitters [16]. It has been suggested that enhanced component-induced responses of detrusor muscle contraction contribute to urinary storage symptoms [17,18,19]. The relationship between detrusor muscle contraction and nocturia and nocturnal polyuria is unclear. However, abnormalities in the detrusor muscle contraction might contribute to nocturia and nocturnal polyuria, one of the urinary storage symptoms. Therefore, examining the changes in the components of detrusor muscle contraction might help elucidate the pathogenesis of urinary storage symptoms. In the present study, we aimed to elucidate the effects of high salt intake on the components of detrusor muscle contraction.

## 2. Materials and Methods

### 2.1. Animals

DR and DS rats (Japan SLC, Shizuoka, Japan) were allowed to take food and water freely, while being kept in a room of controlled temperature and humidity, with alternating light and dark cycles every 12 h. All animal studies were performed following the Guiding Principles for the Care and Use of Laboratory Animals of the Science and International Affairs Bureau of the Japanese Ministry of Education, Culture, Sports, Science and Technology. The study protocols were approved by the Ethics Committee of Nagoya City University (H25-P-11).

### 2.2. Experimental Protocol

First, six-week-old male DR (*n* = 5) and DS (*n* = 5) rats were fed a normal-salt (CE-2 containing 0.3% NaCl, CLEA Japan, Tokyo, Japan) diet for one week to investigate the differences in detrusor muscle contraction characteristics between DR and DS rats. The rats were stabilized for more than 30 min in an incubator (THC-31, Softron, Tokyo, Japan) set at 37 °C, before and after the observation period; their blood pressure was then measured using a noninvasive sphygmomanometer (BP-98A-L, Softron). The rats were euthanized, and the bladder was promptly removed and used for isometric tension study.

Second, six-week-old male DR (*n* = 5) and DS (*n* = 5) rats were fed a high-salt (CE-2 containing 8% NaCl, CLEA Japan) diet for one week to investigate the effects of high salt intake on detrusor muscle contraction. We used 8% NaCl CE-2 to evaluate the dramatic effects of salt intake on bladder function. We set an observation period of one week to prevent the secondary effects of high salt intake; their blood pressure was measured before and after the observation period. The bladder was then promptly removed and used for the isometric tension study and real-time polymerase chain reaction.

### 2.3. Isometric Tension Study

Bladder strips of approximately 3 × 7 mm were prepared. The strips were suspended in an organ bath and located between two platinum electrodes. Subsequently, the organ bath was filled with Krebs solution at 37 °C and bubbled with a gas composed of 95% O_2_ and 5% CO_2_. The tension of the strips was recorded in a Power Lab data acquisition system (ADinstruments, Sydney, Australia) via pressure transducers and analyzed using LabChart 8 (ADinstruments). After an equilibration period of 1 h at 1.0 g tension, the Krebs solution was replaced with KCl 80 mmol/L high K^+^ Krebs solution to contract the strips. Subsequently, the high K^+^ Krebs solution was washed out and the contractile responses to the cumulative administration of carbachol (a muscarinic receptor agonist; 10^−10^ to 10^−4^ mol/L) and electrical field stimulation (EFS; stimulation frequency 1, 2, 4, 8, 16, 32, and 64 Hz, pulse width 5 ms, voltage 160 V, stimulation time 3 s, and main interval 180 s) were measured. The maximum response (E_max_) and the concentration that induced a response of 50% of the E_max_ (EC_50_) of carbachol were calculated by fitting the concentration–response curve to nonlinear regression analysis using Origin version 6.0 (OriginLab Corporation, Northampton, MA, USA). After the measurement of contractile responses to EFS without inhibitors, suramin (a purinergic receptor inhibitor, 10^−4^ mol/L) and atropine (a muscarinic receptor inhibitor, 10^−6^ mol/L) were added sequentially to the organ bath followed by measuring contractile responses to EFS. Among the EFS-induced contractile responses, the components inhibited by suramin and atropine were regarded as purinergic components and muscarinic components, respectively. Each response was evaluated as a ratio to contractile responses to the high K^+^ Krebs solution.

### 2.4. Real-Time Polymerase Chain Reaction

Total ribonucleic acid (RNA) was extracted from excised bladders using the PureLink RNA Mini Kit (Thermo Fisher Scientific, Waltham, MA, USA). Then, complementary deoxyribonucleic acid (cDNA) was synthesized at 42 °C for 20 min and 99 °C for 5 min using a 2720 Thermal Cycler (Applied Biosystems, Foster City, CA, USA). The messenger RNA (mRNA) levels of muscarinic receptors M1, M2, and M3 and those of rho-associated kinases Rock1 and Rock2, nerve growth factor (NGF), and β-actin were measured by real-time polymerase chain reaction (PCR) using the CFX96 Real-time System (Bio-Rad, Hercules, CA, USA). The oligonucleotide sequences of the primers are shown in Table 1. The mRNA expression levels were evaluated as a ratio to the level of β-actin using the 2^−∆∆Ct^ method.

### 2.5. Statistical Analysis

All data were expressed as the mean ± standard deviation (SD). The differences between the DR group mean and the DS group mean were evaluated using the Welch’s *t*-test. *p*-values < 0.05 were considered indicative of statistically significant findings.

## 3. Results

### 3.1. Heart Rate and Blood Pressure in Normal Salt Groups

The heart rate and blood pressure values in the normal salt groups are shown in Table 2. There were no differences in the heart rate or systolic and diastolic blood pressure values between the DR (0.3%) and DS (0.3%) groups, before and after the observation period. There were no differences between the DR (0.3%) and DS (0.3%) groups in the total amount of food (146.0 ± 7.5 and 135.6 ± 7.8 g/body weight, respectively) or the total volume of water (239.3 ± 16.0 and 217.5 ± 9.3 mL/body weight, respectively) consumed over one week.

### 3.2. Isometric Tension Study of the Detrusor Muscle in Normal Salt Groups

There was no difference in the contractile responses to the high K^+^ Krebs solution between the DR (0.3%) and DS (0.3%) groups (data not shown). There were no differences in the contractile responses to the EFS between the DR (0.3%) and DS (0.3%) groups (Figure 1).

There were no differences in the purinergic or cholinergic component-induced responses between the DR (0.3%) and DS (0.3%) groups (Figure 2A,B). The concentration–response curves of carbachol are shown in Figure 3. There were no differences in the E_max_ or the log EC_50_ between the DR (0.3%) and DS (0.3%) groups (Table 3).

### 3.3. Heart Rate and Blood Pressure in High Salt Groups

The heart rate and blood pressure values are shown in Table 4. There were no differences in the heart rate between the DR (8%) and DS (8%) groups. There were no differences in the systolic and diastolic blood pressure values between the DR (8%) and DS (8%) groups before the observation period, while systolic and diastolic blood pressure values were higher in the DS (8%) group than those in the DR (8%) group (*p* < 0.01) after the observation period. There were no differences between the DR (8%) and DS (8%) groups in the total amount of food (142.2 ± 11.5 and 128.4 ± 5.8 g/body weight, respectively) or the total volume of water (805.6 ± 71.1 and 724.2 ± 56.5 mL/body weight, respectively) consumed over one week.

### 3.4. Isometric Tension Study of the Detrusor Muscle in High Salt Groups

There was no difference in the contractile responses to the high K^+^ Krebs solution between the DR (8%) and DS (8%) groups (data not shown). Contractile responses to the EFS were more enhanced in the DS (8%) group than those in the DR (8%) group (*p* < 0.05 for 1, 32, and 64 Hz and *p* < 0.01 for 2, 4, 8, and 16 Hz; Figure 4).

There was no difference in the purinergic component-induced responses between the DR (8%) and DS (8%) groups (Figure 5A). Concurrently, cholinergic component-induced responses were more enhanced in the DS (8%) group than those in the DR (8%) group (*p* < 0.05 for 8, 32, and 64 Hz and *p* < 0.01 for 16 Hz; Figure 5B). The concentration–response curves of carbachol are shown in Figure 6. At a concentration of 10^−6^ mol/L, contractile responses to carbachol were more enhanced in the DS (8%) group than those in the DR (8%) group (*p* < 0.01). There was no difference in E_max_ between the DR (8%) and DS (8%) groups, while the log EC_50_ was lower in the DS (8%) group than that in the DR (8%) group (*p* < 0.05; Table 5). 

### 3.5. Real-Time PCR

The results from real-time PCR reveal no difference in the mRNA expression levels of the muscarinic receptors M1, M2, and M3 or those of the rho-associated kinases Rock1 and Rock2 between the DR (8%) and DS (8%) groups (Figure 7). The mRNA expression level of NGF was higher in the DS (8%) group than that in the DR (8%) group.

## 4. Discussion

Our study demonstrated that there were no differences in the contractile responses of the detrusor muscle to EFS and cumulative administration of carbachol between DR and DS rats fed a normal salt diet; this suggests that there was no difference in detrusor muscle contraction characteristics between DR and DS rats. The volume of drinking water was significantly higher in the high salt than in the normal salt groups. As the volume of drinking water affects bladder function, we could not compare normal to high salt groups. Therefore, we investigated the effects of high salt intake on detrusor muscle contraction using DR and DS rats fed a high salt diet. 

The contractile responses of the detrusor muscle to nerve stimulation by EFS were enhanced in DS rats fed a high salt diet. Detrusor muscle contraction includes purinergic and cholinergic components. Enhanced component-induced responses have been suggested to contribute to urinary storage symptoms because they obstruct bladder extension during urinary storage and decrease bladder capacity [17,18,19]. We further investigated the effects of high salt intake on the components of detrusor muscle contraction, showing that cholinergic component-induced responses of the detrusor muscle were enhanced in DS rats fed a high salt diet. In addition, the concentration–response curves of carbachol, which is a muscarinic receptor agonist, were slightly shifted to the left in DS rats fed a high salt diet, indicating that log EC_50_ was lowered. These results indicate that high salt intake in DS rats makes the detrusor muscle more sensitive to cholinergic stimulation. Therefore, enhanced contractile responses of the detrusor muscle associated with high salt intake may lead to urinary storage symptoms.

To clarify the molecular mechanism underlying the changes in cholinergic pathways due to high salt intake, the mRNA expression levels of cholinergic component-related genes in the bladder were measured. There was no difference in the mRNA expression levels of muscarinic receptors or those of rho-associated kinases, while the mRNA expression level of NGF was increased in DS rats fed high salt diet. NGF has been reported to be involved in hypertrophy of the afferent and efferent neurons in the bladder and to lead to hyperinnervation of the bladder [20,21]. Elevated NGF levels might be involved in the enhancement of cholinergic signals. Future studies are required to verify this hypothesis. The upregulation of NGF is closely related to an overactive bladder [22]. However, the relationship between high salt intake and an overactive bladder has not been extensively studied. From this study, it remains unclear whether upregulation of NGF is directly associated with urinary storage symptoms due to high salt intake.

The relationship between high salt intake and urinary storage symptoms has been evaluated in a few studies. We previously reported that one week of high salt intake in DS rats was associated with upregulated expression of epithelial sodium channel alpha in the bladder epithelium and influenced urinary frequency [12]. Kurokawa et al. reported that twelve weeks of high salt intake in DS rats caused urinary storage dysfunction via increased release of ATP and prostaglandin E2 from distended bladder epithelium [13]. These studies show that high salt intake in DS rats causes abnormalities in the bladder epithelium such as hyperesthesia and increased release of neurotransmitters. However, the effects of high salt intake on the detrusor muscle have not been elucidated. In this study, we are the first to demonstrate that high salt intake in DS rats enhances the detrusor muscle contraction.

High salt intake is known to have various harmful effects, such as hypertension, cardiovascular disease, stroke, renal impairment, osteoporosis, and stomach cancer [23,24]. In this study, systolic and diastolic blood pressure values were higher in the DS rats than those in the DR rats after high salt intake for one week. Being hypertensive for a few days may not change bladder function directly. However, we could not find any evidence supporting this speculation. We showed that one week of high salt intake enhances the cholinergic component-induced responses of detrusor muscle contraction, suggesting that high salt intake adversely affects bladder function even after short-term exposure. Therefore, excessive salt intake should be avoided even for a short period to preserve bladder function. This study has some limitations. First, the study sample size may not be sufficient. Second, the effects of long-term exposure to high salt levels were not investigated. Third, the mechanisms through which high salt intake in DS rats enhances the cholinergic signals of detrusor muscle contraction have not been clarified. Finally, whether these results can be extrapolated to humans remains unclear, as we only used DS rats in this study. Future studies should address these limitations in their design to elucidate the underlying mechanisms.

## 5. Conclusions

Only one week of high salt intake in DS rats enhanced the cholinergic component-induced responses of detrusor muscle contraction via increased sensitivity to cholinergic signals. Enhanced contractile responses of the detrusor muscle may obstruct bladder extension in urinary storage and decrease the bladder capacity. Therefore, excessive salt intake should be avoided to preserve bladder function.

## Figures and Tables

**Figure 1 nutrients-13-00539-f001:**
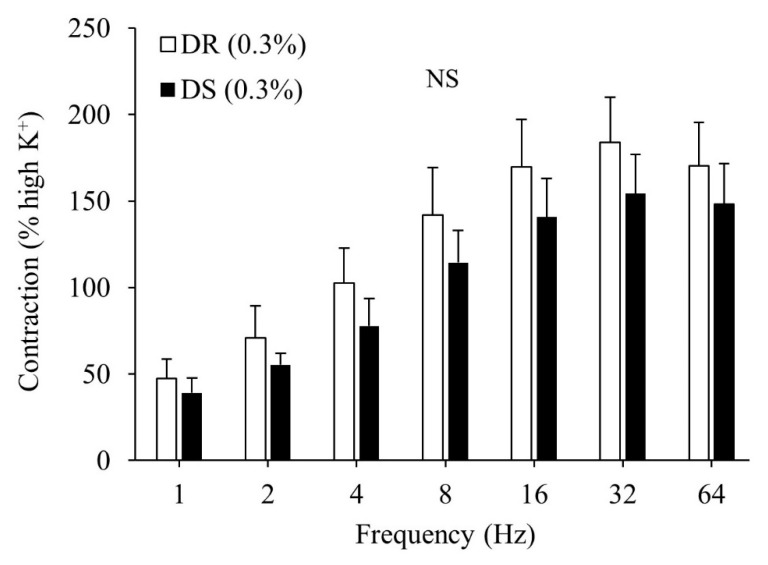
Contractile responses of the detrusor muscle to electrical field stimulation (EFS) in normal salt groups. Data are expressed as mean ± SD, *n* = 5; comparisons were performed using the Welch’s *t*-test. NS: not significant; DR: Dahl salt-resistant; DS: Dahl salt-sensitve.

**Figure 2 nutrients-13-00539-f002:**
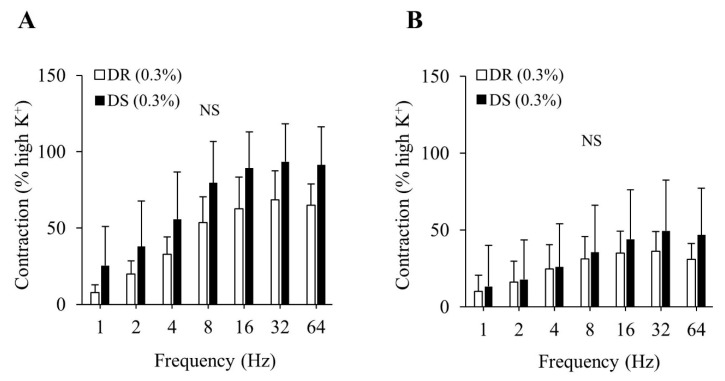
Contractile component-induced responses of the detrusor muscle in normal salt groups. (**A**) Purinergic components, (**B**) Cholinergic components. Data are expressed as the mean ± SD; *n* = 5. Comparisons were performed using the Welch’s *t*-test. NS: not significant; DR: Dahl salt-resistant; DS: Dahl salt-sensitve.

**Figure 3 nutrients-13-00539-f003:**
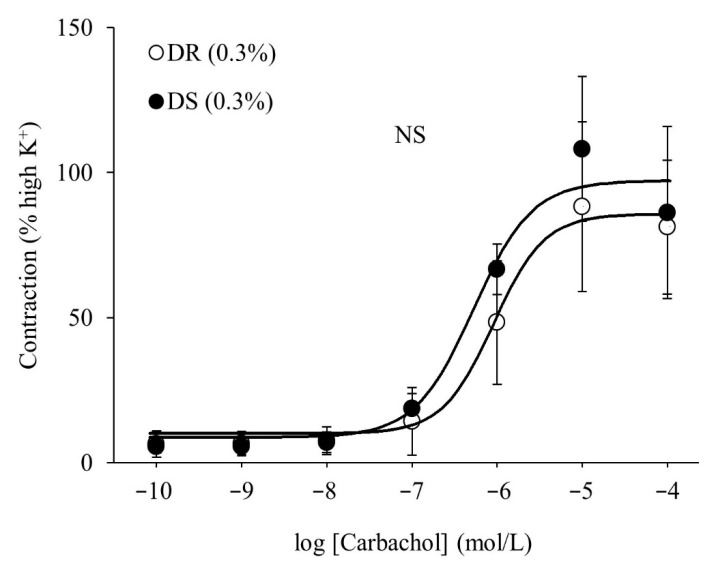
Contractile responses of the detrusor muscle to the cumulative administration of carbachol in normal salt groups. Concentration–response curves of carbachol. Data are expressed as the mean ± SD; *n* = 5. Comparisons were performed using the Welch’s *t*-test. NS: not significant; DR: Dahl salt-resistant; DS: Dahl salt-sensitve.

**Figure 4 nutrients-13-00539-f004:**
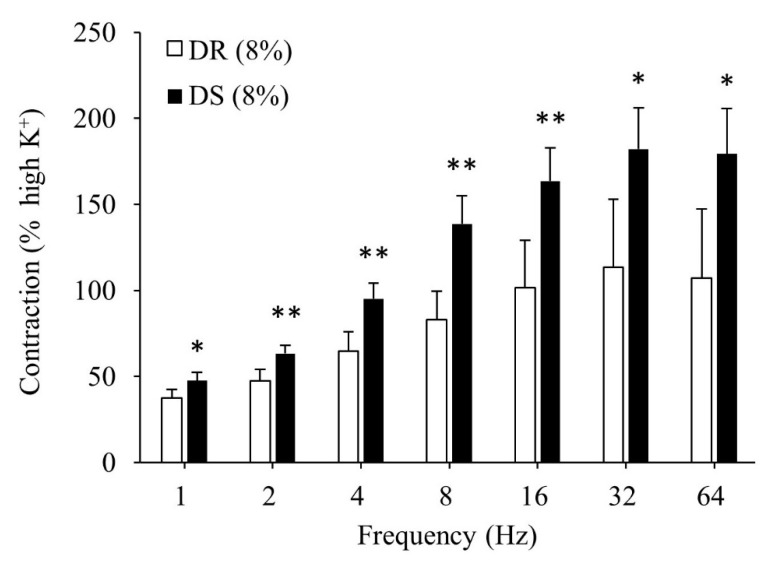
Contractile responses of the detrusor muscle to EFS in high salt groups. Data are expressed as mean ± SD, *n* = 5; comparisons were performed using the Welch’s *t*-test. * *p* < 0.05, ** *p* < 0.01. EFS: electrical field stimulation; DR: Dahl salt-resistant; DS: Dahl salt-sensitve.

**Figure 5 nutrients-13-00539-f005:**
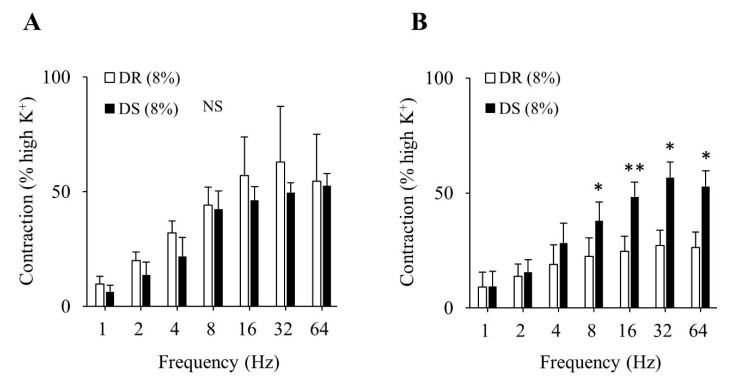
Contractile component-induced responses of the detrusor muscle in high salt groups. (**A**) Purinergic components, (**B**) cholinergic components. Data are expressed as the mean ± SD; *n* = 5. Comparisons were performed using the Welch’s *t*-test. * *p* < 0.05, ** *p* < 0.01. NS: not significant; DR: Dahl salt-resistant; DS: Dahl salt-sensitve.

**Figure 6 nutrients-13-00539-f006:**
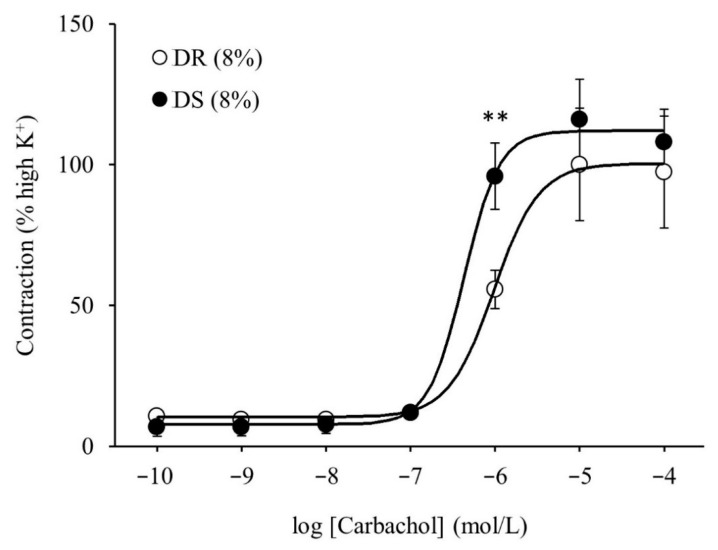
Contractile responses of the detrusor muscle to the cumulative administration of carbachol in high salt groups. Concentration–response curves of carbachol. Data are expressed as the mean ± SD; *n* = 5. Comparisons were performed using the Welch’s *t*-test. ** *p* < 0.01. DR: Dahl salt-resistant; DS: Dahl salt-sensitve.

**Figure 7 nutrients-13-00539-f007:**
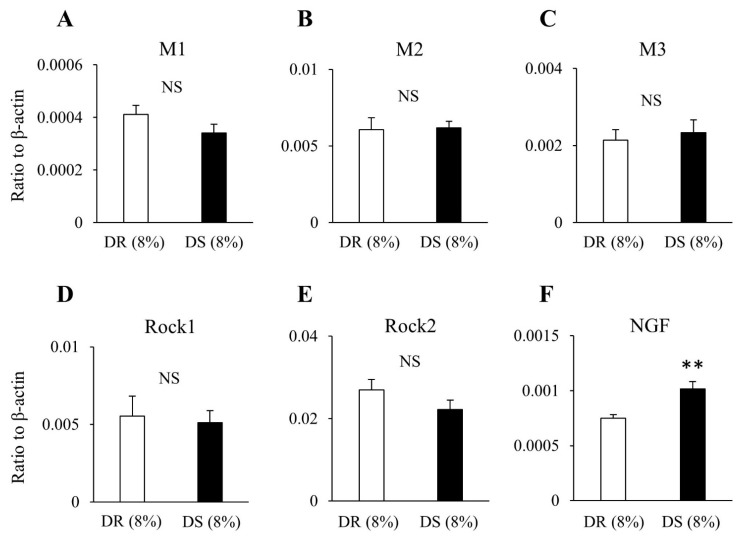
The expression levels of mRNA in the bladder in high salt groups. Data are expressed as the mean ± SD; *n* = 5. Comparisons were performed using the Welch’s *t*-test. ** *p* < 0.01. NS: not significant; (**A**) M1: muscarinic receptor subtype 1; (**B**) M2: muscarinic receptor subtype 2; (**C**) M3: muscarinic receptor subtype 3; (**D**) Rock1: Rho-associated protein kinase 1; (**E**) Rock2: Rho-associated protein kinase 2; (**F**) NGF: nerve growth factor; mRNA: messenger ribonucleic acid; DR: Dahl salt-resistant; DS: Dahl salt-sensitve.

**Table 1 nutrients-13-00539-t001:** Primer sequences.

Gene		Sequence (5′–3′)
*M1*	F	GCACAGGCACCCACCAAGCAG
	R	AGAGCAGCAGCAGGCGGAACG
*M2*	F	TTTGGGACCTGTAGTATGTGACC
	R	GCTTAACTGGGTAGGTCAGAGGT
*M3*	F	TATCCAGCACAGAGAATCCAGAC
	R	GCGCTAACAGTGCAGGAGACAGT
*Rock1*	F	CTGGACATTTGAAGTTAGCCG
	R	CCACTGCTGTATCACATCGTACC
*Rock2*	F	GAACCTACTCCTCGAAGCCG
	R	TGCTTCAGCAGCTCATTCAGTTT
*Ngf*	F	CATGGTACAATCTCCTTCAAC
	R	CCAACCCACACACTGACACTG
*β-actin*	F	TGTGTGGATTGGTGGCTCTATC
	R	CATCGTACTCCTGCTTGCTGATC

**Table 2 nutrients-13-00539-t002:** Heart rate and blood pressure in normal salt groups.

	Initial			Final		
	DR (0.3%)	DS (0.3%)	*p*-Value	DR (0.3%)	DS (0.3%)	*p*-Value
Heart rate (beats per min)	416.7 ± 20.8	434.6 ± 29.8	NS	399.1 ± 25.2	392.8 ± 28.7	NS
Systolic blood pressure (mmHg)	101.9 ± 7.3	114.0 ± 9.3	NS	112.6 ± 5.2	114.8 ± 4.2	NS
Diastolic blood pressure (mmHg)	69.1 ± 12.0	86.8 ± 5.6	NS	81.6 ± 12.4	85.1 ± 5.4	NS

Data are expressed as the mean ± SD, *n* = 5; comparisons were performed using the Welch’s *t*-test. NS: not significant; DR: Dahl salt-resistant; DS: Dahl salt-sensitve.

**Table 3 nutrients-13-00539-t003:** log EC_50_ and E_max_ of carbachol in normal salt groups.

	DR (0.3%)	DS (0.3%)	*p*-Value
log EC_50_ (mol/L)	−6.10 ± 0.31	−6.23 ± 0.29	NS
E_max_ (% high K^+^)	85.7 ± 27.9	98.3 ± 22.6	NS

Data are expressed as mean ± SD; *n* = 5. Comparisons were performed using the Welch’s *t*-test. NS: not significant; DR: Dahl salt-resistant; DS: Dahl salt-sensitve; Emax: the maximum response; EC_50_: a response of 50% of the E_max_.

**Table 4 nutrients-13-00539-t004:** Heart rate and blood pressure in high salt groups.

	Initial			Final		
	DR (8%)	DS (8%)	*p*-Value	DR (8%)	DS (8%)	*p*-Value
Heart rate (beats per min)	415.2 ± 4.9	429.1 ± 18.6	NS	435.0 ± 19.5	410.5 ± 30.7	NS
Systolic blood pressure (mmHg)	102.6 ± 4.5	117.7 ± 13.2	NS	116.3 ± 4.2	142.4 ± 8.5	**
Diastolic blood pressure (mmHg)	78.3 ± 6.6	86.1 ± 12.1	NS	87.7 ± 7.4	100.5 ± 5.5	**

Data are expressed as the mean ± SD, *n* = 5; comparisons were performed using the Welch’s *t*-test. ** *p* < 0.01. NS: not significant; DR: Dahl salt-resistant; DS: Dahl salt-sensitve.

**Table 5 nutrients-13-00539-t005:** log EC_50_ and E_max_ of carbachol in high salt groups.

	DR (8%)	DS (8%)	*p*-Value
log EC_50_ (mol/L)	−6.04 ± 0.05	−6.43 ± 0.12	*
E_max_ (% high K^+^)	99.6 ± 20.4	112.4 ± 9.9	NS

Data are expressed as mean ± SD; *n* = 5. Comparisons were performed using the Welch’s *t*-test. * *p* < 0.05. NS: not significant; DR: Dahl salt-resistant; DS: Dahl salt-sensitve.

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
