# Peer review of "Effects of High Salt Intake on Detrusor Muscle Contraction in Dahl Salt-Sensitive Rats"

_nutrients, 2021, doi:10.3390/nu13020539_

Round 1

Reviewer 1 Report

This manuscript is an investigation the effects of high salt intake on detrusor muscle contraction using rat. This is an interesting paper providing a novel insight into the lower urinary tract status of high salt intake. However, there are some points which have to be clarified and, therefore, the manuscript should be revised according to the following comments.

Introduction

The author reported high salt intake is one of the causes for urinary storage symptoms. This is true, however in this study, the author focused on the detrusor muscle function. Matsuo et al. reported that salt intake induces nocturia and nocturnal polyuria. The author has to describe the connection between nocturia and abnormalities in detrusor muscle contraction in the introduction section.

Materials and Methods

How to decide the duration of high salt intake? The purpose of this study is “change” of bladder muscle function and RNA. Generally, lifestyle modification is not affecting our many symptoms only 1 week. Please clarify this point.

Results

In this study, systolic and diastolic blood pressure values were higher in the DS group than those in the DR group. The author describes the bladder function and blood pressure change in human study. In my opinion, few days blood pressure could not change bladder function directly. Please discussion this point more in detail.

Moreover, the mechanism of high salt intake and storage symptoms should be described in discussion section from previous study.

Discussion

About the NGF. In the current study, only NGF is increased, this is related to overactive bladder for many animal. Is this true, the patient of high salt intake has overactive bladder? If so, the author describes more from previous study.

Author Response

Reviewer 1

Thank you for the reviewer’s constructive comments. We believe that these comments will make our manuscript better.

  1. Introduction

The author reported high salt intake is one of the causes for urinary storage symptoms. This is true, however in this study, the author focused on the detrusor muscle function. Matsuo et al. reported that salt intake induces nocturia and nocturnal polyuria. The author has to describe the connection between nocturia and abnormalities in detrusor muscle contraction in the introduction section.

Response 1

Thank you for your suggestion. In the revised manuscript, we have included more details to indicate the connection between nocturia and abnormalities in detrusor muscle contraction. It has been suggested that enhanced component-induced responses of detrusor muscle contraction contribute to urinary storage symptoms [17-19]. The relationship between the detrusor muscle contraction and nocturia and nocturnal polyuria is unclear. However, abnormalities in the detrusor muscle contraction might contribute to nocturia and nocturnal polyuria, one of the urinary storage symptoms (p.1 lines 39-40, p.2 lines 55-57).

  1. Materials and Methods

How to decide the duration of high salt intake? The purpose of this study is “change” of bladder muscle function and RNA. Generally, lifestyle modification is not affecting our many symptoms only 1 week. Please clarify this point.

Response 2

Thank you for your comment. We used CE-2 containing 8% NaCl to evaluate the dramatic effects of salt intake on bladder function. We set an observation period of 1 week to avoid the secondary effects of high salt intake. In humans, as you have indicated, the effect of lifestyle modification may not be realized in one week. Therefore, further studies on the impact of long-term consumption of high salt levels are warranted. We have incorporated this information in the materials and methods and discussion sections of the revised manuscript (p.2 lines 71-73, p.7 lines 206-207).

  1. Results

In this study, systolic and diastolic blood pressure values were higher in the DS group than those in the DR group. The author describes the bladder function and blood pressure change in human study. In my opinion, few days blood pressure could not change bladder function directly. Please discussion this point more in detail.

Response 3

We agree with your opinion. As you have indicated, being hypertensive for a few days may not change bladder function directly. However, we could not find any evidence supporting this speculation. We have reflected this comment in the discussion section of the revised manuscript (p.7 lines 199-202).

  1. Moreover, the mechanism of high salt intake and storage symptoms should be described in discussion section from previous study.

Response 4

Thank you for your suggestion. We have added more information in the discussion section to describe the mechanism underlying the relationship between high salt intake and urine storage symptoms (p. 7 lines 188-197).

  1. Discussion

About the NGF. In the current study, only NGF is increased, this is related to overactive bladder for many animal. Is this true, the patient of high salt intake has overactive bladder? If so, the author describes more from previous study.

Response 5

Thank you for your suggestion. As you have indicated, the upregulation of NGF is closely related to overactive bladder. However, the relationship between high salt intake and overactive bladder has not been studied. In this study, it is unclear whether upregulation of NGF is directly associated with urinary storage symptoms due to high salt intake.

Reviewer 2 Report

  1. The study would have been complete if a second salt-sensitive group had received a salt-free diet as opposed to the salt-rich diet group.
  2. Also, there are no data for baseline comparisons of the DR vs DS group
  3. And finally, isn’t it somewhat expected to have a stronger reaction to a salt-enriched diet in the salt-sensitive than in the salt-resistant animals?
  4. The study sample size is marginal for sound results and statistical comparisons
  5. Otherwise, the manuscript is very well written and comprehensive

Author Response

Reviewer 2

Thank you for the reviewer’s constructive comments. We believe that these comments will make our manuscript better.

  1. The study would have been complete if a second salt-sensitive group had received a salt-free diet as opposed to the salt-rich diet group.

Response 1

Thank you for your suggestion. We previously investigated the effects of normal-salt (0.3% NaCl) and high-salt (8% NaCl) diet on the bladder function in salt-resistant and salt-sensitive rats. As a result, the amount of drinking water was significantly higher in the high-salt group than in the normal-salt group. As the amount of drinking water affects bladder function, we could not compare the normal-salt group to the high-salt groups [12]. Therefore, we only used high-salt diet groups.

  1. Also, there are no data for baseline comparisons of the DR vs DS group

Response 2

Thank you for your comment. Before the observation period, there were no differences in the systolic or diastolic blood pressure values between the DR and DS groups. We have added the data and description in the revised manuscript (p.3 lines 116-118, p.4 table 2).

  1. And finally, isn’t it somewhat expected to have a stronger reaction to a salt-enriched diet in the salt-sensitive than in the salt-resistant animals?

Response 3

We agree with you. We expected the salt-sensitive rats consuming a salt-enriched diet to show enhanced detrusor muscle contraction. However, it has been unclear whether the detrusor muscle contraction is affected and how the contractile components change. In this study, we demonstrated that a salt-enriched diet in salt-sensitive rats enhanced the cholinergic component of the detrusor muscle.

  1. The study sample size is marginal for sound results and statistical comparisons

Response 4

We agree with you. The sample size in this study was n=5 in each group and might be marginal for valid results. We have included this issue in the revised manuscript as a limitation of the study (p. 7, line 206).

Round 2

Reviewer 1 Report

In my previous review comments and author's response,

"Response 5
Thank you for your suggestion. As you have indicated, the upregulation of NGF is closely related to overactive bladder. However, the relationship between high salt intake and overactive bladder has not been studied. In this study, it is unclear whether upregulation of NGF is directly associated with urinary storage symptoms due to high salt intake."

The author should add about the OAB and BGF comments in discussion section.

Author Response

Thank you for the careful review of our manuscript, and the constructive comments provided. We believe that these comments have helped us considerably in improving our manuscript.

  1. In my previous review comments and author's response,

"Response 5

Thank you for your suggestion. As you have indicated, the upregulation of NGF is closely related to overactive bladder. However, the relationship between high salt intake and overactive bladder has not been studied. In this study, it is unclear whether upregulation of NGF is directly associated with urinary storage symptoms due to high salt intake."

The author should add about the OAB and BGF comments in discussion section.

Response 1

Thank you for your suggestion. We have incorporated the mentioned information in the discussion section of the revised manuscript, as suggested (p. 10 lines 244-247).

Reviewer 2 Report

Thank you for addressing this reviewer's comments.

Regarding the response to Query 2, I feel that it is inadequately addressed, as the authors only present baseline data on blood pressure, but not on the basic experiment concerning the contractile response characteristics of the detrusor muscle. One cannot conclude that there is a change if baseline data are missing. Please address.

Also, in association with Query 1 response, do the authors have an explanation why the DS rats did not have increased fluid intake compared to the DR rats?

Author Response

Thank you for the careful review of our manuscript, and the constructive comments provided. We believe that these comments have helped us considerably in improving our manuscript.

  1. Regarding the response to Query 2, I feel that it is inadequately addressed, as the authors only present baseline data on blood pressure, but not on the basic experiment concerning the contractile response characteristics of the detrusor muscle. One cannot conclude that there is a change if baseline data are missing. Please address.

Response 1

We appreciate your observations. We have additional data regarding the blood pressure and contractile responses of the detrusor muscle in DR and DS rats fed a normal salt diet for one week. We found that there were no differences between DR and DS rats fed a normal salt diet, in terms of the contractile responses of the detrusor muscle to EFS and cumulative administration of carbachol; this suggests that there were no differences in detrusor muscle contraction characteristics between DR and DS rats. We have added new tables 2 and 3 (p.4) and figures 1, 2, and 3 (p.4 and 5), and have incorporated this information in the Materials and Methods, Results, and Discussion sections of the revised manuscript (p.2 lines 70-83, p.3-5 lines 122-158, p.9 lines 218-224).

  1. Also, in association with Query 1 response, do the authors have an explanation why the DS rats did not have increased fluid intake compared to the DR rats?

Response 2

We appreciate your pertinent question. Both DR and DS groups demonstrated obvious increases in fluid intake (8% vs. 0.3% each). However, we are unable to explain the differences in fluid intake between DR and DS rats.